# Using B cell receptor lineage structures to predict affinity

**Duncan K. Ralph** *, **Frederick A. Matsen IV**

Fred Hutchinson Cancer Research Center, Seattle, Washington, USA

* dralph@fredhutch.org

**Data Availability Statement:** https://doi.org/10.5281/zenodo.3728068.

**Funding:** This research was supported by National Institutes of Health grants R01 GM113246, R01

## Abstract

We are frequently faced with a large collection of antibodies, and want to select those with highest affinity for their cognate antigen. When developing a first-line therapeutic for a novel pathogen, for instance, we might look for such antibodies in patients that have recovered. There exist effective experimental methods of accomplishing this, such as cell sorting and baiting; however they are time consuming and expensive. Next generation sequencing of B cell receptor (BCR) repertoires offers an additional source of sequences that could be tapped if we had a reliable method of selecting those coding for the best antibodies. In this paper we introduce a method that uses evolutionary information from the family of related sequences that share a naive ancestor to predict the affinity of each resulting antibody for its antigen. When combined with information on the identity of the antigen, this method should provide a source of effective new antibodies. We also introduce a method for a related task: given an antibody of interest and its inferred ancestral lineage, which branches in the tree are likely to harbor key affinity-increasing mutations? We evaluate the performance of these methods on a wide variety of simulated samples, as well as two real data samples. These methods are implemented as part of continuing development of the `partis` BCR inference package, available at https://github.com/psathyrella/partis.

Comments

Please post comments or questions on this paper as new issues at https://git.io/Jvxkn.

## Author summary

Antibodies form part of the adaptive immune response, and are critical to both naturally acquired immunity and vaccine response. Next generation sequencing of the B cell receptor (BCR) repertoire provides a broad and highly informative view of the DNA sequences from which antibodies arise. In many cases we would like to identify which of these BCR sequences correspond to antibodies with the highest affinity for a particular antigen. Existing experimental methods of selecting antibodies are effective, but time-consuming and expensive. In this paper we introduce new computational methods that use evolutionary information from the family of related BCR sequences to predict the affinity of each resulting antibody for its corresponding foreign antigen. When combined with information on the identity of this antigen (which we do not attempt to predict), these methods

AI146028, R01 AI138709, R01 AI120961, and U19 AI128914 (FAM). The research of FAM was supported in part by a Faculty Scholar grant from the Howard Hughes Medical Institute and the Simons Foundation. The research of DKR was supported by a 2018-2019 University of Washington/Fred Hutch Center for AIDS Research Young Investigator award (P30 AI027757). https://www.nih.gov/ https://hhmi.org https://www.simonsfoundation.org/ http://depts.washington.edu/cfar/ The funders had no role in study design, data collection and analysis, decision to publish, or preparation of the manuscript.

**Competing interests:** The authors have declared that no competing interests exist.

should provide a source of effective new antibodies that can then be experimentally synthesized and tested for function.

This is a *PLOS Computational Biology* Methods paper.

## Introduction

Antibodies are the foundation of both vaccine-induced immunity and many important therapeutics. They stem from B cells through the processes of VDJ rearrangement and somatic hypermutation (SHM), which yield a vast repertoire of B cell receptors (BCRs) within each person. Each clonal family begins from a single naive B cell that has encountered its cognate antigen, which then reproduces in a germinal center (GC), diversifying and evolving as SHM drives affinity maturation. We can probe the BCR repertoire via next generation sequencing (NGS), and with computational methods divide it into groups of clonally-related sequences [1] stemming from the same rearrangement event (clonal families).

Of the many sequences in the BCR repertoire, we are generally interested in those that code for the highest affinity antibodies. There exist several effective experimental methods of finding high affinity B cells. Both cell sorting and antigen baiting have been used to find a variety of important antibodies [2–4]. These approaches have several challenges. For instance, constructing a stable baiting antigen for certain conformational epitopes can be challenging. However their main limitation is that they require large investments of time and resources.

Deep sequencing of the BCR repertoire yields tens of thousands to millions of sequences from this same pool, some of which undoubtedly correspond to high affinity antibodies. And in many cases, this NGS data already exists, since it is frequently collected as a matter of course when studying, for instance, antibodies isolated using the cell sorting approach. If we had some way of identifying a handful of these sequences that are likely to correspond to high affinity antibodies, then they could be synthesized and tested for their binding properties. This could yield a rich source of novel antibodies.

In most practical cases we would like to choose only antibodies that are effective against a particular pathogen or epitope. Because the evolutionary signatures in sequence data can tell us much more about affinity than they can about specificity, we separate this task into two parts: finding antibodies or families with the desired specificity, and finding antibodies with high affinity regardless of specificity. While de novo epitope prediction from sequence data alone is a rapidly developing field, the challenges are still such that specificity determination is best accomplished with non-sequence-based information, such as cell sorting or information about vaccine challenge. Because our expertise is solely in analyzing sequence data, the methods in this paper thus deal only with affinity prediction. However, to give an idea of what is possible, we also review techniques that have been used to enrich for particular specificities (see Discussion).

In our experience, there are two common use cases for methods of choosing antibodies from NGS data, depending on what prior information is available. If we have a previously-isolated antibody of interest, then specificity has already been determined and we want to choose only among the sequences in that antibody's family. In the absence of such an antibody, we are

instead choosing from among many different families in the repertoire, and our affinity prediction methods must be paired with some method of enriching for families with the desired specificity. While the second case is more difficult, because both affinity and specificity vary between families, it also holds the promise of much greater rewards, since novel antibodies from previously unknown families could bind with much higher affinity, or to new epitopes.

In this paper we first test, on both simulated and real data samples, a variety of methods to choose individual high-affinity sequences both from within single families, and from among all families of a given specificity in the repertoire. We find that an observed sequence's similarity to its family's amino acid consensus sequence (measured by hamming distance, and abbreviated `aa-cdist`) is a highly effective predictor of affinity in both of these cases. Because the other, more poorly performing metrics provide independent information not in `aa-cdist`, we also combined them using several machine learning approaches. We were not, however, successful in training such a method that substantially outperformed `aa-cdist`, which we believe is likely due to several unique features of the combined tree and sequence spaces.

We next introduce an entirely new metric to predict the change in affinity along each branch of a family's inferred phylogenetic tree (abbreviated Δ-affinity). This new metric, called local branching ratio (`nuc-lbr`), draws on ideas from [5] and [6]. It is designed for situations where we have an antibody of interest, and want to know which branches along its inferred lineage are likely to harbor the most important mutations.

Since we focus on B cells we desire to predict affinity, which in our simulations we define as the inverse of the dissociation constant in a biochemically-motivated model (see Methods). However the metrics we test are more directly measuring evolutionary fitness (the expected number of offspring), and are thus also of much wider applicability. For instance, while writing this paper we discovered that some of the authors of [5] had concurrently found that `aa-cdist` is predictive of fitness in the context of viral evolution, and also derived a mathematical proof that `aa-cdist` is in some circumstances the optimal metric [7]. We would also expect `nuc-lbr` to predict fitness-increasing mutations in an inferred viral lineage just as effectively as it predicts affinity increases for antibodies.

These metrics are essentially evolutionary in nature, and thus require at least a handful of sequences from each clonal family. They require nucleotide sequences only, with no structural information. All of these metrics can be calculated with simple options in the freely available `partis` software package https://github.com/psathyrella/partis, which also performs BCR annotation, simulation, clonal family clustering, and per-sample germline inference [1, 8, 9]. However, we note that `aa-cdist` is simple enough that for users who have already grouped their sequences into clonal families, a simple sequence alignment GUI may suffice.

## Results

### Affinity and fitness

The metrics that we use to predict affinity and Δ-affinity for each sequence begin by considering that sequence in the context of its clonal family of related sequences. Whether this context is a phylogenetic tree or a sequence alignment, it provides information about the evolutionary history that led to the sequence's development. The metrics use this information to find the sequences stemming from cells that were fittest, in the evolutionary sense, during the GC reaction. Because a key purpose of the GC reaction is to apply selective pressure in order to direct evolution toward antibodies that bind to pathogens with high affinity, fitness is generally correlated with affinity. It is important to note, however, that this fitness also depends on other factors, such as the ability to recruit T cell help. We thus generally refer to "predicting affinity" in this paper, but it should be understood that the metrics are more directly connected to GC

reaction fitness, and the extent to which this fitness correlates to affinity must be judged on a case by case basis, particularly in real data. In simulation, we define specific mathematical connections between each sequence's affinity and its expected number of offspring. The two real data samples in this paper, in contrast, measure neutralization concentration rather than affinity, and while these quantities are generally expected to correlate, we cannot say to what extent in any specific instance.

## Metrics

We measured the ability of a variety of metrics to predict affinity and Δ-affinity, all of which are summarized in Table 1. Each is also described in the following paragraphs.

As mentioned above, each observed sequence's hamming distance to its family's amino acid consensus (abbreviated **`aa-cdist`**) is the best predictor of affinity (where smaller distance is better). While this metric has been used in an ad hoc way [10], to our knowledge the only attempt to measure its performance was [11], which tested on only a single, small family in the supplemental information. In many cases the calculated consensus sequence itself is not observed, and it is possible that in such cases this unobserved consensus would be a better predictor than the nearest observed sequence, but we have not evaluated this (see Discussion). When calculating the consensus, ties are treated as ambiguous positions, which are then ignored in the hamming distance.

Nucleotide local branching index (abbreviated **`nuc-lbi`**) uses the family's phylogenetic tree to measure the "branchiness" in the local area around each node [5]. This branchiness is calculated by integrating the total branch length surrounding the node with a decaying exponential weighting factor (S1 Fig). While `nuc-lbi` performs worse than `aa-cdist` in almost all regions of parameter space, it is nevertheless important because it contains some independent information, provides an obvious path to improving `aa-cdist`, and serves as the basis for `nuc-lbr`. Our implementation of `nuc-lbi` includes several modifications to the original formulation. We perform an independent optimization of the $\tau$ locality parameter, although we arrive at a comparable final value. We also introduce a normalization scheme, which although amounting only to a switch to human-interpretable units, is important because without it `nuc-lbi` calculated in different papers with different $\tau$ values cannot be compared.

We also introduce a new version of local branching index that incorporates only nonsynonymous mutations (abbreviated **`aa-lbi`**), which significantly outperforms the original nucleotide version in all regions of parameter space. It in fact outperforms `aa-cdist` in some

**Table 1. Metrics used in this paper to predict affinity (top) and Δ-affinity (bottom).** Those marked with a * perform much better than the others, and are recommended for all practical use. The sign of the metric's correlation with the predicted quantity is indicated by "± corr.".

| affinity metrics | shorthand | ± corr. | description |
|---|---|---|---|
| AA consensus distance * | `aa-cdist` | - | hamming distance to the clonal family's amino acid consensus sequence |
| nuc. local branching index | `nuc-lbi` | + | total branch length in the local area of the nucleotide tree [5] |
| AA local branching index | `aa-lbi` | + | `nuc-lbi` calculated on a tree reflecting only nonsynonymous mutations |
| somatic hypermutations | `n-shm` | + | number of nucleotide somatic hypermutations |
| nucleotide consensus distance | `nuc-cdist` | - | nucleotide version of `aa-cdist` |
| **Δ-affinity metrics** | shorthand | ± corr. | description |
| nuc. local branching ratio * | `nuc-lbr` | + | ratio of total branch length in the (nucleotide) tree below the node to above the node |
| change in nuc-lbi | `Δ-nuc-lbi` | + | change in local branching index compared to the parent node |
| AA local branching ratio | `aa-lbr` | + | `nuc-lbr` calculated on a tree reflecting only nonsynonymous mutations |

regions; however (like nuc-lbi) its poor performance with low selection strength and when choosing among all families recommend against its use as a standalone metric.

Because affinity maturation via somatic hypermutation should be, in the long run, an affinity-increasing process, many papers have used the number of somatic mutations as a proxy for affinity (abbreviated **n-shm**, and distinguished from our abbreviation SHM for the process). Unfortunately in practice this metric performs very poorly because it chooses leaves, which harbor many novel mutations. Novel mutations are in general overwhelmingly deleterious, and in leaves have not yet been evaluated by significant selective pressure. The underlying idea that affinity increases with distance from root is valid; however it is necessary to go some way back toward root from the leaves (indeed this is what aa-cdist accomplishes).

In addition to aa-cdist, we also show results for its nucleotide analog **nuc-cdist**. This is useful mainly as a way to understand the importance of information from the amino acid translation table, and thus differences between aa-cdist and nuc-lbi.

An additional metric that we do not evaluate is the multiplicity of each unique sequence. It can be experimentally challenging to disentangle underlying cell numbers from other factors such as primer bias and varying expression levels [12–14]. Thus while higher multiplicity is sometimes used as a standalone metric to predict higher fitness [10], we do not evaluate it here because many data sets that we encounter do not include the measures necessary for an accurate estimation. For samples that are known to have reliable multiplicity information, however, this can be passed to partis (see https://git.io/JJCGe), and our implementations of nuc-lbi, aa-lbi, and aa-cdist include extensions such that it will be properly accounted for in the calculations (see Methods).

To predict affinity-increasing mutations (Δ-affinity) we mainly focus on the local branching ratio **nuc-lbr**. This novel metric uses the same branch length integrals as nuc-lbi, but instead of summing in all directions, it compares the branchiness among the node's offspring to that of its parents and siblings (S1 Fig). This ratio of branchiness below vs above the node thus quantifies the possibility that an affinity-increasing mutation occurred along the branch immediately above the node, since such a mutation would increase fitness among the node's offspring.

In order to provide some baseline for the effectiveness of nuc-lbr, we also evaluate the change in nuc-lbi from the parent node (abbreviated **Δ-nuc-lbi**) as a predictor of Δ-affinity. While this functions adequately, it is always significantly worse than nuc-lbr.

As for nuc-lbi, we also introduce an amino acid version of nuc-lbr (abbreviated **aa-lbr**); however it does not perform better than the standard version.

## Evaluation framework

The first step toward confidence in any method is measuring its performance in all corners of its parameter space. For the metrics in this paper, that space is constructed from sequences and trees and is vast, complicated, and high-dimensional. In order to measure performance, we began with a previously-described simulation framework [15], extending it to allow a more comprehensive variation of parameters, and rewriting to optimize for speed. We then performed scans across all reasonably plausible values of parameters that could affect performance.

The simulation begins by constructing a naive sequence via VDJ rearrangement. It then chooses a number of "target" sequences at some distance from the naive, each of which represents a potential optimal antibody. Repeated rounds of SHM then apply selective pressure to direct the naive sequence's offspring toward these targets.

We measure performance using a metric called "top quartile accuracy gap" that we believe emphasizes the practical use case for these metrics: choosing a handful of sequences from deep sequencing data in order to invest substantial resources into synthesis and binding evaluation. While these metrics predict affinity, and are thus correlated with it, quantifying overall correlation with something like the Pearson coefficient would not measure effectiveness for our use case. This is because we only care about the very small fraction of sequences at the highest affinity values, whereas measures of overall correlation count all sequences equally and are thus dominated by low- and medium-affinity sequences. Furthermore, correlation implicitly values sensitivity and specificity equally, whereas for us specificity is much more important: wasting real-world experimental resources on a bad antibody is usually worse than missing a few of the best binders.

To measure performance we instead imagine that we have chosen the top few sequences according to the predictive metric, and then ask whether they are also among the top few in affinity. We quantify this by taking the average of their affinity quantiles. We then compare this mean quantile to that of a hypothetical perfect method that simply ranks sequences by their true affinity. The amount by which the mean affinity quantile of sequences chosen by the actual metric deviates from that of the perfect metric, further averaged over quantiles from 75 to 100 (i.e. choosing between 25% and 0% of sequences), forms the basis of our performance evaluation. Since this is, roughly speaking, the gap between the affinity of the top quartile of sequences chosen by the metric vs by a hypothetical perfect metric, we refer to it as the "top quartile accuracy gap" or "accuracy gap". When predicting Δ-affinity rather than absolute affinity, we use a similar procedure, except that "quantiles" is replaced by "N ancestors", the number of branches in the tree that separate the true and inferred affinity-increasing mutations (see Methods).

We supplement these comprehensive simulation scans with validation results on two small real data samples.

## Simulation results

We first show simulation performance for a single parameter scan for both affinity and Δ-affinity prediction (Figs 1 and 2). This scan varies observation time (in units of N generations) from 50 to 3000 while holding other variables constant. This corresponds roughly to varying the mean frequency of somatic nucleotide mutations among observed sequences from 2% to 25%.

We then show analogous scans for a variety of other parameters when choosing within families (S2 Fig; the corresponding among-families plots may be found at https://zenodo.org/record/3929565). These scans correspond to varying the most important simulation parameters: longitudinal sampling, carrying capacity, number of sampled sequences, sequence-to-affinity mapping, and selection strength. Generalizing across these scans, we find that aa-cdist performs consistently better than most other metrics, typically at 5-10% from perfect. aa-cdist's performance is largely recapitulated, and sometimes slightly exceeded, by aa-lbi. aa-lbi frequently has a slight advantage when choosing within a family and at very early observation times, while aa-cdist is often better choosing among families and at long times. However their behavior diverges as we vary selection strength (S2 Fig bottom right), with aa-lbi doing slightly better for high selection strength, but dramatically worse near neutral evolution. Also of note, nuc-cdist performs similarly to, but worse than, nuc-lbi in many of the scans. This is consistent with the hypothesis that under high selection strength the local branching calculation does a better job of quantifying branchiness/evolutionary density than does consensus distance, but that this advantage is overwhelmed by the large benefit

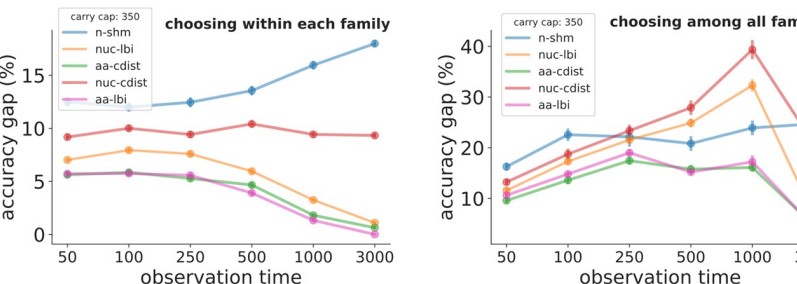

**Fig 1. Simulation performance for affinity prediction vs. observation time when choosing sequences within each family (left) and among all families of a given specificity in the repertoire (right) for the metrics in Table 1.** The y-axis is a percentage describing how much each metric deviates from the best possible performance when choosing the top few sequences (see Evaluation framework section in text). For example an "accuracy gap" value of 5% for `aa-cdist` would mean that if you decided to choose, say, the top 7% of sequences using `aa-cdist` that the resulting sequences would on average be in the top 12% of affinity. An observation time of 50 (3000) generations corresponds to a nucleotide SHM frequency of roughly 2% (25%). The longer simulation times should be thought of as a series of GC reaction/reentry cycles, as would for instance be seen during anti-HIV bNAb development over several years. Each point shows the mean ± standard error of 30 samples, where each sample consists of 1500 sequences from 10 families. Similar plots across ranges of other simulation parameters can be found in S2, S3, S4 and S5 Figs, and at https://zenodo.org/record/3929565.

to including amino acid information in `aa-cdist`. Under low selection strength, on the other hand, there is little for the local branching calculation to measure. Meanwhile `n-shm` performs much worse than the other metrics in almost all situations. We also show performance for a variety of different models of cell export from the GC (S3 Fig), as well as the effect of varying the number of "target sequences" (the simulation's representation of hypothetical ideal antibodies, S4 Fig).

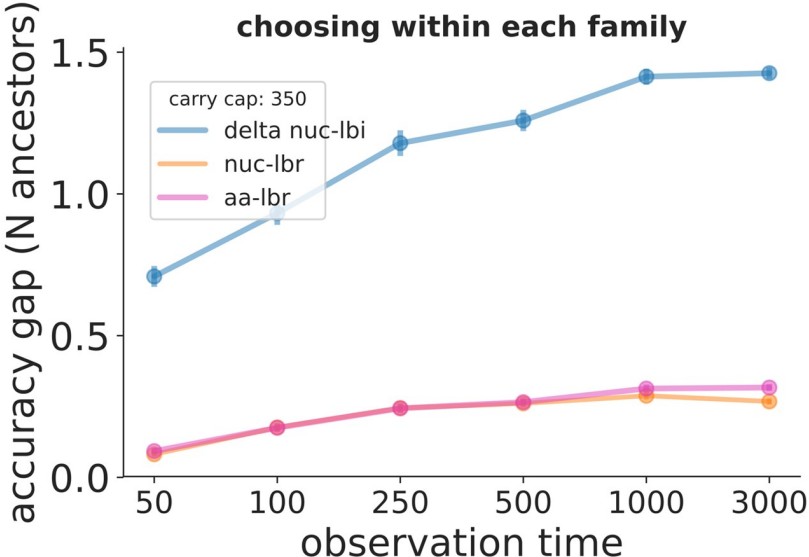

**Fig 2. Simulation performance for Δ-affinity prediction vs. observation time for the metrics in Table 1.** The y-axis describes how close each metric comes to achieving the best possible performance. Here we are predicting the location of affinity-increasing mutations in a lineage of inferred ancestral sequences, and for example an "accuracy gap" value of 0.5 for `nuc-lbr` would mean that if we choose the best inferred ancestral sequence using `nuc-lbr` that we would, on average, be 0.5 branches away from a branch containing an actual affinity-increasing mutation. See Fig 1 caption for details. Similar plots across ranges of other simulation parameters can be found at https://zenodo.org/record/3929565.

In order to provide a more stringent test of performance when choosing among all families, we also show results on samples where (unlike those described above) parameters vary between the families in each sample (S5 Fig). We first observe that, as in Fig 1, all methods perform better when choosing within families (left column) than among them (right column). To evaluate the effects of adding between-family parameter variance on choosing among families (right column), we focus on whether the vertical spread of values left of the dashed line encompasses the values to its right. In other words, the points left of the dashed line tell us the effect of changing the mean values of the parameters (but with no between-families variance), while those right of it tell us the effect of adding variance between families. The biggest effect of adding this variance is that `n-shm` performs much worse when observation time (i.e. % SHM) varies between families (right plot in third row), however it also appears that `aa-cdist` degrades by about 5% when selection strength varies between families (right plot in second row). These surprisingly moderate effects are likely because families vary widely in their final characteristics even when they start with the same parameter values, since almost all important characteristics are determined by the very small (typically 3–5), and thus highly stochastic, number of beneficial mutations that occurs in each family.

In the $\Delta$-affinity prediction (Fig 2) `nuc-lbr` performs much better than `$\Delta$-nuc-lbi`, on average identifying a branch that is 0.1–0.2 steps in the tree from the branch that actually has an affinity-increasing mutation. This can be thought of as choosing the correct branch eight or nine out of ten times, and being one branch away the other one or two times. `aa-lbr`, on the other hand, performs similarly to `nuc-lbr`, i.e. in contrast to (absolute) affinity prediction, the addition of amino acid information does not improve performance. The $\Delta$-affinity performance plots for the remaining parameter scans are at https://zenodo.org/record/3929565; they show `nuc-lbr` typically between 0.1 and 0.5 steps from perfect, and consistently much better than `$\Delta$-nuc-lbi` (but similar to `aa-lbr`).

Many other variations of these parameter scans, and plots comparing multiple slices for each metric, can be found at https://zenodo.org/record/3929565.

## Data results

Validation on real sequence data serves a different, complementary purpose to validation on simulation. Real data validation is more stringent, and thus more useful, in the sense that by definition it perfectly recapitulates the properties of real data. But on the other hand real data is less stringent, and thus less useful, because it is more difficult to produce and thus is only ever available for a very restricted set of parameter values. For instance looking at one real data sample might only explore large, high-mutation trees with strong selection, but ignore performance at other parameter values. This means that designing any method using real data validation alone is extremely risky, since it only provides information about how the method performs for the particular combination of parameters in those data sets. In the present case, because affinity is so expensive to measure, real data has a further problem: while there are many papers with both NGS data and affinity information (e.g. [16–31]), to our knowledge very few of them both measure affinity for more than a handful of sequences per clonal family and have made that data public. This is, nevertheless, certainly better than nothing, and can provide confidence that nothing has gone egregiously wrong in the simulation studies.

Here we have chosen data from two well-known papers on anti-HIV antibodies [32, 33]. There are several aspects of HIV that, a priori, we expected would make it a very challenging environment for these metrics. First, these papers both measure $IC_{50}$ neutralization concentrations rather than affinity. To handle this we simply use $1/IC_{50}$ in place of affinity, and hope that neutralization and affinity are correlated enough for the metrics to work. Second, HIV's

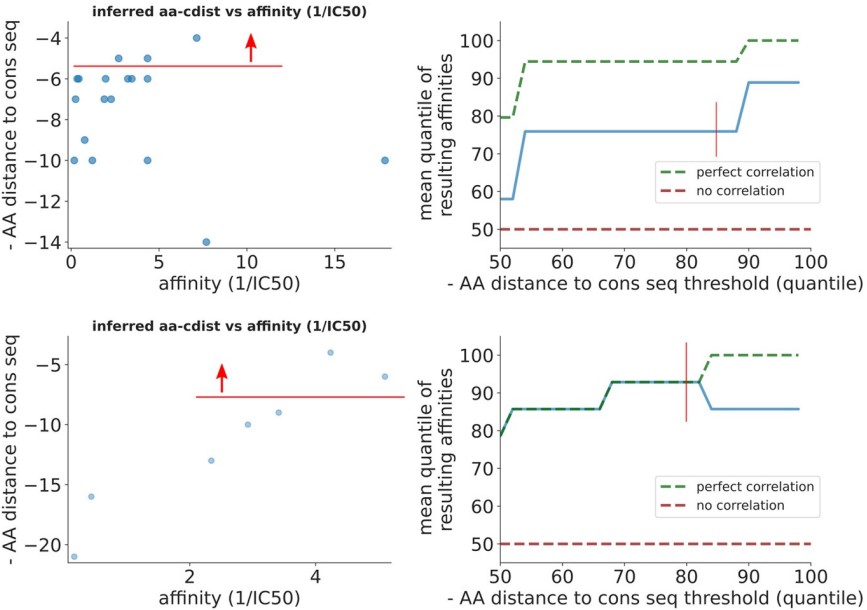

**Fig 3. Performance on real data for `aa-cdist` from [32] (top) and [33] (bottom).** The scatter plots (left) show the raw correlation between `aa-cdist` and measured "affinity" (here actually $1/IC_{50}$), while the quantile plots (right) show the relative affinity ranking for sequences chosen using `aa-cdist` (blue) compared to perfect (green) and random chance (red). For instance, an x value of 85 on the top right plot corresponds to 75 on the y axis, meaning that if we chose the top 15% of sequences using `aa-cdist` that these sequences would on average rank in the top 25% by affinity (thin red vertical lines). Another way of viewing this is that, at x of 85, the blue line is about 20% below the green line, meaning performance is 20% worse than perfect; this interpretation corresponds to the "accuracy gap" values in Fig 1. We also show the equivalent plots for `nuc-lbi` (S6 Fig).

vast diversity both globally and within each subject means that the viruses that applied selective pressure during each antibody's development were in all likelihood quite different from the viruses against which the antibodies' neutralization strengths were tested. These papers also report the geometric mean $IC_{50}$ over many viruses from the three main global HIV clades, and for the sake of simplicity we use these overall values for validation. As for almost all current work, these papers also use unpaired sequencing, which not only ignores any signal from light chain evolution, but also typically means measuring $IC_{50}$ values for chimeric antibodies with non-native light chain sequences. Finally, we note that these are the only real data samples on which we have run our methods—it would of course have been uninformative to run on many different samples before deciding which to include in this paper.

With the final reminder that these are extremely small sample sizes, we find that on real data both `aa-cdist` (Fig 3) and `nuc-lbi` (S6 Fig) perform roughly in line with expectations from simulation. We find that `aa-cdist` is generally between 0 and 20 quantiles from perfect, while `nuc-lbi` ranges from 10 to 30.

## Discussion

We have demonstrated effective, practical methods for two tasks common to the analysis of B cell receptor deep sequencing data. First, we find that a sequence's similarity to its clonal family's amino acid consensus (`aa-cdist`) is an excellent predictor of that sequence's affinity, and is a highly effective way to choose a handful of sequences for synthesis and testing. Second, we find that a new metric that we construct called local branching ratio (`nuc-lbr`) is similarly effective for the related task of predicting which branches in a single lineage are likely to

harbor affinity-increasing mutations (i.e. predicting Δ-affinity). These metrics are both implemented in the existing `partis` annotation and clonal family inference package, available at https://github.com/psathyrella/partis. By testing performance on simulation generated with a wide variety of parameters, we showed that choosing the best few sequences according to these metrics will likely result in antibodies that are also among the highest affinity. We further find that the number of somatic hypermutations (`n-shm`), while a frequently-used heuristic affinity predictor [10], performs very poorly. We also showed performance on two very small real data samples, which provide a confidence-boosting cross check. We emphasize that in these methods we make no attempt at specificity prediction, since this is better achieved through non-sequence-based methods such as clinical information (e.g. use of vaccine challenge). Thus in cases where no previously-isolated antibody is available to restrict consideration to a single family, while `aa-cdist` is effective at choosing the highest affinity antibodies from among many families, it is necessary to pair it with a non-sequence-based method of enriching for families with the desired specificity. Many examples of such methods are described below (see Previous work).

There are many avenues for future improvement to these methods. We currently use the plain consensus, which by considering each site independently ignores information about how sequences evolved together at different sites. By taking a single family-wide consensus, it also does a relatively poor job of handling families with several widely-separated branches. While its performance does not drop significantly in such cases (this is tested by the variety of target sequence configurations in S4 Fig), there is likely room for improvement. When presented with widely-separated branches, the consensus calculation will in many cases simply ignore all but the largest branch. This is not ideal, since when evolving toward separate targets, the less-numerous branches might hold a significant number of effective antibodies. We could address this by adding locality to the consensus calculation, perhaps by dividing the tree into sublineages and then calculating the consensus for each sublineage [34].

We instead introduced `aa-lbi` as an initial effort to combine `aa-cdist` and `nuc-lbi`, since it provided a simpler path to implementation. The addition of amino acid information indeed dramatically improves performance: `aa-lbi` does much better than `nuc-lbi` almost everywhere, in fact largely recapitulating the performance of `aa-cdist`. It does not, however, perform well in near-neutral evolutionary scenarios, which are likely a common feature of real data [35]. Thus although `aa-lbi` is promising, we do not recommend it for general use at this time.

Another disadvantage of `aa-lbi` is that the necessary phylogenetic tree and ancestral sequence inference add additional uncertainty and computation time. The results presented here benchmark `aa-lbi` on the true tree and ancestral sequences, rather than the noisy reconstruction one would obtain given real data. Although (nucleotide) `nuc-lbi` performs to expectation on small real data samples (S6 Fig), and we show that `nuc-lbi` appears largely insensitive to poor phylogenetic inference (S8 Fig), we have not quantified the effect of this inference uncertainty on performance, and in some cases it could be significant. For `aa-cdist`, on the other hand, while the consensus of a small number of sampled sequences is not a perfect predictor of the full-family consensus (S7 Fig), there is no inference uncertainty inherent to the (very simple) consensus calculation.

Towards a different future direction, we acknowledge that the assumption that all sites are equivalent is particularly inaccurate for BCR sequences: not only are the complementarity determining regions (CDRs) more important for binding than framework regions (FWKs), but activation-induced deaminase (AID) activity results in strongly context-dependent mutation patterns. This information could be incorporated into `aa-cdist` when calculating the hamming distance by assigning different weights to different regions, perhaps using [36] or

[37]. For instance, hypothetically, perhaps a one amino acid difference in the CDR should count for twice (or half) as much as a difference in FWK. Similarly, perhaps mutations at highly mutable positions should count less toward the distance than those at less mutable positions. This approach could be further improved by incorporating structural and functional information, for instance using deep mutational scanning data such as [38] to develop models of binding change upon mutation.

Another limitation of these metrics is that they do not incorporate information from insertion or deletion mutations (SHM indels). Because SHM indels can have a large impact on function in the (relatively rare) cases where they occur, this is a significant limitation. Since indels are not generally treated as informative characters by tree inference programs, this limits the potential utility of adding SHM indel information to the tree-based metrics in the near future. It would probably be possible, however, to design a new consensus calculation that incorporates indels into `aa-cdist`; however we have not experimented with this approach.

While `aa-cdist` performs well on its own, there is clearly significant independent information in the remaining metrics. We thus invested a large effort in developing a machine learning approach incorporating many different tree and sequence variables (Table 3). We found the best performance with a decision tree regression (abbreviated `dtr`), but were unable to significantly and consistently improve on the performance of `aa-cdist` alone. We believe this failure stems from two factors. First, it is possible that there isn't that much additional information in the other variables: `aa-cdist` is in many regions already quite close to perfect. Second, because the relative performance of different metrics varies dramatically between different parts of parameter space, the `dtr` has to make a very accurate determination of its location in this space before deciding how to use the input variables. In making this determination, however, it is limited to inferred variables, which provide only a tenuous link to true parameter space. To take one example, the relative usefulness of `nuc-lbi` and `n-shm` completely reverses between low and high selection strength (blue and orange in bottom right of S2 Fig). While the `dtr` input variable Fay/Wu H does predict selection strength [35], it is far too noisy to give the `dtr` an accurate idea where along the x axis it is for a given family (results not shown). We thus find that the `dtr` generally recapitulates the performance of the best single metric at each point in parameter space, but is rarely much better, and sometimes a bit worse. Since the best single metric is usually `aa-cdist`, and `aa-cdist` is both vastly simpler to calculate and interpret and is not subject to even the possibility of overtraining, this means that `aa-cdist` is a better choice overall.

Finally, while it is clear that the observed sequence that falls closest to the family consensus is likely of high affinity, we have not tested whether the actual consensus sequence (when it is not observed) would be even better. The selection of such an unobserved consensus sequence for synthesis would be risky, since unlike observed sequences, there would be little direct evidence showing that it produces a stable, functional antibody. However, it is possible that this would be a good strategy in cases where the overriding goal is finding the antibody with the highest possible affinity, and where synthesizing one extra sequence is not a large burden.

Shifting now to predicting affinity-increasing mutations, `aa-lbr`'s performance is largely identical to that of `nuc-lbr`, showing that in contrast to predicting absolute affinity, the incorporation of amino acid information does not improve performance. We speculate that this could be because the high resolution inherent to the numerator/denominator distinction that allows use of a very large $\tau$ (see Methods) means that `nuc-lbr` already sees far enough up and down the tree to average out synonymous mutations. Another avenue for improvement could be drawing more explicitly on the analogy to absolute affinity prediction, such as using the change (from parent node) in hamming distance to a local amino acid consensus sequence.

An additional concern with `nuc-lbr` is that because it is sensitive to the detailed long range ancestral lineage structure, it requires more accurate phylogenetic inference than `nuc-lbi` (S8 Fig). While `nuc-lbi` performs well even with the very heuristic (but also very fast) trees that `partis` makes by default, `nuc-lbr` would benefit from the more sophisticated inference provided by external packages such as linearham [39] or RAxML [40]. Because they also provide ancestral sequence inference, these programs will in any case usually be required for `nuc-lbr`, since unless internal branches are short enough to contain only a few mutations, a prediction of which branches contain important mutations is not very meaningful.

## Previous work

Because the metrics we have presented do not provide information on specificity, we first review prior work on ways to enrich for antibodies that are specific to a particular pathogen or epitope. These use information from sources beyond the NGS data, and will provide a key component of most practical workflows involving our methods. As described above, cell sorting and baiting provide a very direct way to identify antigen-specific antibodies, and thus when combined with `partis` seed sequence clustering, also antigen-specific families in the NGS data (see [41] for an example of this workflow). Yeast display can be used for high-throughput specificity and affinity screening of natively paired heavy and light chains [42]. A very recent paper uses a microfluidics-based approach to experimentally evaluate antibody specificity in high throughput [43] (of particular relevance to our results, they find no correlation between `n-shm` and measured affinity). Another recent paper clusters sequences into specificity groups using amino acid hamming distance on a subset of CDR residues deemed likely to contribute to binding [44]. These paratope residues are chosen using a deep learning approach trained on a large database of existing structural data [45]. This is a particularly exciting complement to our affinity prediction methods, since given an antibody of interest it would allow consideration not just of that antibody's family, but also of unrelated families with similar paratopes (and thus specificities).

A less-direct way to enrich for a particular specificity is to apply a large immune stimulus, such as vaccination, and then limit the analysis to plasma cells from families that expand around 7 days after vaccination [46–52]. In model organisms, it is also possible to cut out tissues where antigen-specific B cells are likely to concentrate (such as lymph nodes or Peyer's patches). With longitudinal sampling, a family's persistence over time can be a strong indicator of specificity in the presence of either chronic infection [53] or the application of multiple vaccinations [49]. With several subjects that have been exposed to the same antigen, we can select shared lineages either using simple sequence similarity [46–48, 54] or a Bayesian mixture model incorporating also clonal abundance [55]. With an outside source of antigen-specific sequences (e.g. from cell sorting or public databases), we can choose sequences that are similar [50].

Specificity prediction without any of these extra information sources, i.e. from sequence data alone, is much more difficult. It would involve de novo structure and binding prediction, which are not currently practical, although much recent work focuses on these problems [56–60].

Many papers have focused on finding families that have been subject to strong selection, which when applied to the BCR repertoire is then assumed to correlate to families with higher affinity. Probably the most common approach to picking highly selected families has been to measure the ratio of nonsynonymous to synonymous mutations. Observed nucleotide mutations are separated into synonymous (S) and nonsynonymous (NS), the ratio NS/S is calculated, and values much larger (smaller) than one indicate positive (negative) selection. A

number of corrections [61–64] and optimizations [65] have also been developed to reduce its dependence on the baseline mutation model.

In perhaps the earliest B cell specific effort to pick families [66], the authors calculate a fairly comprehensive variety of simple tree shape metrics on a handful of small trees (e.g. number of nodes, root-tip distance, outgoing degree, trunk length). While some correlations were noted, it was largely a descriptive exercise without strong quantitative conclusions. Work several years later [67] found correlations between some of the same metrics and GC reaction parameters in a mathematical model. However, a later paper [63] noted that neither [66] nor [67] tested under realistic conditions: both assumed 100% sampling depth, i.e. included every cell from the GC reaction history in the final tree. After accounting for realistic sampling, this later paper found that the simple tree metrics lost their predictive power. They then devised a new metric combining a small amount of tree information with NS/S by ignoring terminal mutations (which as described above are much less likely to lead to high affinity antibodies). This is analogous to a combination of `nuc-lbi`, which uses tree structure to measure selection, with `aa-cdist`, which reduces the influence of tip mutations. A more recent paper [35] found that NS/S is higher in CDR than FWK regions, and calculated the likelihood-based fitness from [5] for several trees, but found no significant relationships between changes in this fitness and NS/S or CDR vs FWK.

A more complex statistic known as Fay/Wu H [68] has also been used in the context of BCR repertoires. It quantifies any excess at high values of the site frequency spectrum, and can be thought of as measuring the amount of shared mutation in the family, or equivalently the prevalence of selective sweeps. It was used in [35] to determine that families that expand rapidly in response to vaccination are generally positively selected. We independently verified that it indeed identifies highly-selected families (results not shown), and thus included it in the `dtr`.

A metric called the log offspring number ratio, which provided a starting point for `nuc-lbr`, was introduced in [6]. This metric looks in the tree for pairs of sibling branches where one branch has a mutation and the other does not. It is then calculated as the log of the ratio between the number of offspring in the mutated vs non-mutated branches. The distribution of this value is then calculated separately for NS and S mutations, and a rightward (leftward) shift in the NS distribution indicates positive (negative) selection. It unfortunately is rendered less useful by several issues. First, it counts offspring all the way down the tree, so that ancestors get credit for fitness improvements in all of their offspring, so it cannot be used to find the location of important mutations. It also ignores sibling pairs in which either both edges have mutations, or either branch has multiple mutations, which together can amount to throwing out a large fraction of the tree. Finally, it can only detect in-progress (i.e. not-yet-fixed) incremental selection, and the NS and S mutation rates must be of the same order of magnitude.

The remainder of previous work has focused on choosing single sequences from within a family, and separates into experimental and theoretical studies. Experimental papers typically first choose a family based on the methods above for specificity to the pathogen of interest, then exclude sequences with "bad" features (e.g. highly hydrophobic, or with free cysteines or atypical indels), and then rank the remaining sequences by measures such as `n-shm`, V gene usage, and CDR3 length. On the theoretical side, `nuc-lbi` has undoubtedly enjoyed the most practical use. Introduced in [5], it was originally a quick heuristic replacement for a more complex likelihood-based metric. Its ability to predict fitness in real influenza data has been shown in [69], and it features prominently on the nextstrain [70] web site https://nextstrain. org/flu/seasonal/h3n2/ha/2y?c=lbi.

In the only work we're aware of that evaluated `aa-cdist`, the authors adapted an approach from structural studies called maximum entropy [11]. This models the multiple

sequence alignment of a family as a multivariate gaussian, and then takes probability (i.e. near-ness to a gaussian's peak) as a correlate of affinity. This maximum entropy metric performs well in their tests, however its generality is unknown, since it was trained and tested using only one relatively small real data sample. Although they avoid statistical overtraining by training and testing on disjoint parts of the sample, a single sample can only tell us about performance at the one point in parameter space at which it resides. Furthermore, since training consists of optimizing to that particular sample, other regions of parameter space are by construction very likely to have worse performance. Finally, the resulting software does not seem to be pub-licly available, although the underlying gaussian modeling framework is available at https://github.com/carlobaldassi/GaussDCA.jl. Nevertheless, as a cross check in the supplemental information when predicting binding vs non-binding antibodies, they use area under curve (AUC) to compare their method (0.97) to `aa-cdist` (0.86). This indicates the surprising use-fulness of `aa-cdist`, especially since `aa-cdist` is certainly not subject to the same poten-tial for overtraining.

## Methods

### Simulation framework

The simulation of each clonal family in this paper begins with a naive sequence generated by the `partis` simulation command [8]. Since in this paper we focus on affinity maturation rather than VDJ rearrangement, we refer to that paper for all details on its implementation and validation. This naive sequence is then passed to the `bcr-phylo` package [15] for GC reaction simulation. For this paper we have extended the original software by adding a number of new parameters to allow for more comprehensive variation, as well as optimizing for speed. The full simulation of a family combining `partis` and `bcr-phylo` can be run with the fol-lowing script: https://git.io/JvFcW.

The `bcr-phylo` simulation begins by generating a number of "target" nucleotide sequences from the naive nucleotide sequence. These targets represent hypothetical optimal antibodies toward which evolution will be directed, and are chosen at fixed distance from the naive sequence (default 15 amino acid hamming distance). The representation of the GC reac-tion proceeds generation by generation, beginning with the naive sequence's single cell. In each generation, a number of offspring is chosen for each cell from a Poisson distribution with parameter λ determined by that cell's affinity; if a cell has zero offspring in a generation its line-age ends. The correspondence between nucleotide sequence and affinity is by default deter-mined by amino acid hamming distance to target sequence. For a detailed description refer to the supplement of [15], but the gist is that at each generation a limiting amount of antigen is apportioned among the cells based on their affinity (inverse dissociation constant) using equa-tions of chemical binding equilibrium, and each cell's λ is calculated based on its acquired anti-gen and the carrying capacity. The net result is a monotonic increase in mean number of offspring as a cell's sequence draws closer to a target sequence. We also tested a model with a much less discrete distribution of possible affinity values, where the distance for each amino acid pair is rescaled by their BLOSUM similarity (S2 Fig, bottom left). In order to introduce a selection strength parameter (bottom right of S2 Fig), instead of determining the Poisson parameter directly from affinity, we smear it out (rescale it) by drawing from a normal distri-bution. The normal distribution's mean and variance are calculated such that for a selection strength of 0, affinity has no effect on offspring number, while for selection strength 1 the Pois-son parameter is determined directly by affinity (but since the number of offspring is still drawn from that Poisson, there is still significant stochasticity). Specifically, the parameters for the $i^{th}$ cell's normal distribution are given by $\mu_i = \bar{\lambda} + s(\lambda_i - \bar{\lambda})$ and $\sigma_i = (1 - s)\sigma_\lambda$, where $s$ is

the selection strength, $\lambda_i$ is the $i^{th}$ cell's unrescaled (affinity-determined) Poisson parameter, and $\bar{\lambda}$ is the mean and $\sigma_\lambda$ the standard deviation of unrescaled Poisson parameters of all cells. Note that $\sigma_i$ is somewhat arbitrary, but this choice ensures that the variance in rescaled $\lambda$ over cells is comparable for all values of $s$. The simulation is terminated after some number of generations, referred to as the observation time.

When each child cell is generated, its nucleotide sequence has some probability to suffer a point mutation. This probability is set based on experimental values, but averages roughly one point mutation per generation. Synonymous mutations have no effect on affinity, while a non-synonymous mutation that decreases the distance (whether plain or BLOSUM-scaled hamming) to a target sequence increases affinity, and thus mean number of offspring (Poisson parameter). While context-dependent mutation is implemented in `bcr-phylo` (using the S5F model [71]), because this feature increases run times by too much to allow the generation of the large samples needed for our validation, all samples shown here have this option turned off. In the limited sample sizes that we have run with context dependence turned on (results not shown), neither `aa-cdist` nor `nuc-lbi` consistently performed either better or worse. The main pattern was that `n-shm` performed significantly worse with context dependence turned on.

Given the complete simulated tree of cells, we have to decide which we will sample. In order to simplify the many ways of choosing N cells from M generations, we focus on two cases that cover two alternative models for GC cell export: either sampling a fraction of the desired number of cells every few generations (top row of S2 Fig), or sampling all cells at the end (all others). In addition, in order to mimic typical phylogenetic programs that infer ancestral sequences, we then also recursively sample all MRCA sequences starting with the set of sampled cells (although we have also run extensive validation without this option, results not shown, and the only significant effect is due to the change in the fraction of leaf sequences). Different models of GC export also stipulate different levels of bias toward exporting higher affinity cells. We attempt to cover the various possibilities with three options (S3 Fig): uniform random (default), sampling with probability proportional to affinity, and sampling the cells in order of perfectly decreasing affinity.

Unless otherwise noted, in order to isolate the effect of the value of each parameter, every family in a sample has the same parameter value. However, to demonstrate the effectiveness of choosing among all families of a given specificity requires that we also vary parameters between families in a sample. In S5 Fig we show the effects of increasing the variance of parameters between families by first measuring performance on samples where every family has the same parameter value (points left of dashed lines), and then on samples where the value for each family is chosen at random (right of dashed lines in S5 Fig). The idea is to first give some idea of the effect of changing only the mean value of the parameter (left of the dashed line), and then of changing the variance (right of the dashed line). For the samples right of the dashed line, the value for each family is drawn at random from the choices listed in Table 2 (which is summarized on the figure's x axis as a range).

An unavoidable feature of our approach is that we must simulate a vastly larger number of sequences than we want to sample. Since every cell in the evolutionary history of the family must be simulated, and we want to decide which ancestral cells to sample at the end, large carrying capacities and observation times dramatically increase the required time and memory. For example, each point in Fig 1 uses only $4.5 \times 10^4$ final sampled sequences, but requires actually simulating $3 \times 10^8$ sequences. And the biggest `dtr` training sample, with $2.5 \times 10^6$ final sampled sequences, required simulating roughly $7 \times 10^{10}$ sequences (Table 4).

**Table 2. Parameter variance choices for samples used in points to right of dashed lines in S5 Fig.** "Fig Row" refers to the row in S5 Fig. The indicated parameter for each family is drawn at random from the listed values, and the "first", "second", and "third" columns refer to the respective x values to the right of the dashed line in the Figure.

| Fig Row | parameter | first | second | third |
|---|---|---|---|---|
| 1 | N sampled | 25, 50, 75 | 15, 50, 150, 500 | |
| 2 | selection strength | 0.5, 0.75, 1.0 | 0.25, 0.67, 1.0 | 0.1, 0.25, 0.75, 1.0 |
| 3 | observation time | 50, 100, 200 | 50, 150, 500, 1000 | |
| 4 | N sampled | 50, 150, 300 | 25, 150, 500 | |
| 4 | selection strength | 0.5, 0.75, 1.0 | 0.25, 0.75, 1.0 | |
| 4 | observation time | 100, 250, 500 | 50, 250, 1000 | |

When we simulate families with high SHM frequencies, for simplicity we treat them as a single long-running GC reaction rather than multiple sequential reactions. Real antibodies with 20-30% SHM are generally assumed to have undergone many cycles of GC completion and reentry over a number of years (although antibodies with 20% SHM have been observed after only 28 days [72]). Treating this instead as a single long-running GC reaction, however, is probably a good approximation, because GC completion followed by reentry of memory B cells is similar to a strong selective sweep [73]. This means that GC completion and reentry is largely equivalent to increasing either the observation time or selection strength.

While it seems likely that most families in most real data correspond to regions of parameter space far from any optimal antibody or target sequence, it is nonetheless important to explore behavior as a family reaches the target sequence toward which it has been evolving. For the initial target distances and carrying capacities used in our simulations, this corresponds to observation times greater than several thousand generations (i.e. SHM around 20%). In order to maintain some diversity in these cases, we introduce a minimum target distance threshold (which we set to two amino acid changes) below which affinity does not increase. Thus when sequences draw nearer to a target sequence than this threshold, they no longer experience selective pressure to move closer to the target sequence.

Another limitation of our current approach is that while it can model a single family evolving toward multiple target sequences, it cannot model competition between families. Doing this explicitly would be computationally prohibitive, since all the families in a repertoire would need to interact with each other. However the important features could likely be mimicked by allowing each family's carrying capacity to vary with time, simulating the effects of other families' over- or under-utilization of available resources. The biggest problem stemming from this current limitation is that each family operates under a fixed carrying capacity, so we cannot evaluate the effectiveness of using clonal family size as a method of choosing high-affinity families. As a practical matter, however, this may not be a significant problem, since the extent to which family size is an effective predictor depends entirely on how much competition between families really occurs, which has not yet been definitively established. If different families are in perfect competition (for instance if we believe that there are either many families within each GC, or lots of transport between different GCs), then family size is by definition that family's fitness. On the other hand if there is little between-families competition, family size would tell us little about fitness, being determined instead by random chance (e.g. by which family happened to develop in the most well-resourced GC).

While we have great confidence that our simulation framework effectively recreates the important features and variation of BCR evolution, we would prefer to also validate on a completely independent package. Unfortunately, while there are many packages that come close to simulating what we need [31, 74–76], to our knowledge they all lack necessary features.

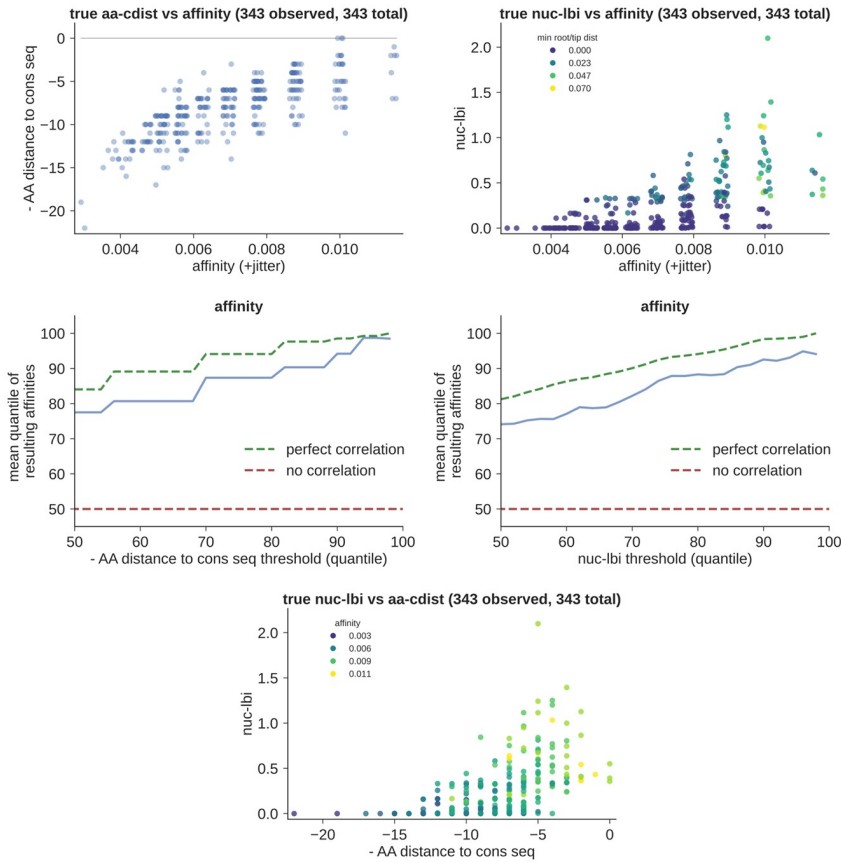

**Fig 4. Simulation performance of `aa-cdist` (left) and `nuc-lbi` (right) on a single, representative family.** First showing scatter plots of each metric vs affinity (top), and then the quantile performance plots (middle, see text and Fig 1 for explanation). The `nuc-lbi` scatter plot is colored by the distance to the "edge" of the tree, i.e. minimum distance to either tip or root: `nuc-lbi` is less accurate (and biased low) for nodes near the edge of the tree (darker color). At bottom, we show the correlation between `aa-cdist` and `nuc-lbi`, colored by affinity.

Most frequently, they do not simulate complete nucleotide sequences, and instead draw a cell's affinity either directly from some distribution, or based on a few key residues.

## Evaluation framework

While we described the basics of our performance evaluation above, there are many steps between a simple metric vs affinity scatter plot and the "top quartile accuracy gap" values in Fig 1. The question we're trying to answer is, if we take the best few sequences according to our metric, how high can we expect those sequences to rank in affinity? In a metric vs affinity scatter plot (left side of Fig 3, or top row of Fig 4) this means taking the highest few points by y, and seeing where they rank in affinity (x). We calculate these rankings as the quantile in affinity, averaged over the chosen sequences. For instance the sequences in the top 4% of affinity have quantiles from 96%-100%, and thus mean quantile 98%. The top 4% of sequences according to some metric, on the other hand, might have affinity quantiles spread between 85%-100%, which could give a mean of 92%. This example both neglects ties and assumes infinite sample size; in practice both are important, which results in jumps and horizontal lines as in Fig 3. This is one reason why it's important to compare to the mean quantile of a hypothetical perfect metric (green dashed lines), rather than to a constant value of 100. We plot this

resulting mean affinity quantile for thresholds from the 50th to 100th quantile (i.e. choosing from half to none of the sequences, right side of Fig 3, or middle row of Fig 4).

In order to compare performance over many different parameter choices, we need to summarize this plot with one number. We do this by defining the deviation from perfect as the difference between the metric's performance and that of a hypothetical perfect method (distance between blue and green lines). We then average this quantity from the 75th to 100th quantiles (i.e. choosing varying fractions of the top quartile). This average, reported as a mean ± standard error over many statistically independent samples, is what appears in Fig 1 as "top quartile accuracy gap" or "accuracy gap".

When predicting Δ-affinity, we cannot simply report the mean Δ-affinity quantile of the chosen sequences because we want to account for being close to, but not exactly on, the correct branch. We instead imagine moving upward on the tree from the node of interest until reaching a branch containing an affinity-increasing mutation. We report the number of steps (i.e. ancestors) that were necessary, so if we're exactly right this number is 0. Searching only upward reflects the fact that a mutation can only affect the fitness of nodes below it, and thus a high `nuc-lbr` value at a node immediately above an important mutation is likely due to random chance rather than a signal of selection. Nodes with high `nuc-lbr` that are several steps below such a mutation, on the other hand, simply reflect the fact that increased fitness typically takes several generations to manifest itself as an increase in observed offspring. In other words searching downward would improve the apparent performance of a metric, but only by counting as successes cases that were successfully predicted only through random chance. Another reason we do not also search in the downward direction is that in a practical sense it is much more useful to know that the important mutation is above a node than below it. We could imagine in the lab testing one or a few branches above a node, but because of the bifurcating nature of trees there would be far too many potential branches below (not to mention adding the ambiguity of potentially going up and then down, i.e. how to count cousins). One potential issue with this step-counting approach is that it gives equal credit for being off by long and short branches. We thus also performed extensive validation using the total branch length traversed, rather than number of steps (results not shown). The performance was generally similar, and is also probably less relevant since during inference we don't control how long the branches are. For instance any metric would appear to do worse on sparsely-sampled trees with long branches.

## Distance to family consensus sequence (`aa-cdist` and `nuc-cdist`)

One nice feature of consensus distance metrics is that, unlike `nuc-lbi` and `nuc-lbr`, there is no inference inaccuracy: they are a direct result from observed sequences. However, these observed sequences will in some cases not accurately represent the entire family. In order to quantify this inaccuracy, we calculated the full-family consensus sequence, and then compared it to consensus sequences calculated with smaller subsets of the family (S7 Fig). While the nucleotide consensus can be quite inaccurate (top), we only calculate it in order to inform comparison between `aa-cdist` and `nuc-lbi`. The amino acid consensus, on the other hand, is quite accurate, reaching an error of one position (out of around 130) only for sampled families smaller than 10 sequences and very early times (i.e. when almost no selection has yet occurred, S7 Fig bottom left).

## local branching index (`nuc-lbi`)

The authors of [5] introduce `nuc-lbi` as an approximate metric to supplement their more complex likelihood-based measure. However, they find that the two perform very similarly, so

the full likelihood calculation is probably more useful for building intuition (and motivating `nuc-lbi`) than for practical use. The `nuc-lbi` score for each node in the tree is computed as the total of all nearby branch lengths, weighted by an exponential function of the distance away from that location with scale $\tau$ (S1 Fig).

**$\tau$ optimization.** The decay length $\tau$ determines the size of the local area that impacts each node's `nuc-lbi`. The authors of [5] suggest using 0.0625 times the average pairwise distance between sequences (elsewhere they use the $T_C/\sqrt{\log N}$ and $T_C/15$, for $T_C$ the coalescence time, but since we're not using coalescent models this is less useful here). While these estimates are based on a thorough optimization, they result in a $\tau$ that depends on which sequences are sampled, which precludes comparisons between families, as well as between samples in different papers. Since these comparisons are important for us, we perform an optimization from a somewhat different perspective, although we end up with a comparable value.

We begin by noting that the sequence length has a profound effect on the distances over which trees branch, and for BCR sequences it is constant across families and samples. The minimum possible branch length, corresponding to one point mutation, is equal to the inverse sequence length $1/\ell_{\text{seq}}$. Thus from first principles/dimensional analysis we expected this to be a reasonable guess for $\tau$. However, we also want a value that gives optimal performance, so we measured performance vs $\tau$ in a number of parameter scans similar to Fig 1. The strongest dependence was vs number of sampled sequences as a fraction of carrying capacity (S9 Fig, top row). While the most important message from this plot is probably that `nuc-lbi` performance does not depend very strongly on $\tau$, there is a pronounced peak in performance at $1/\ell_{\text{seq}} \simeq 1/400 = 0.0025$, especially when choosing within families (top right), where `nuc-lbi` is most important. We get a similar value if we (roughly) calculate the recommendation from [5] for 10% SHM: $0.0625 \times 0.10 = 0.00625$. Thus we recommend using $\tau = 1/\ell_{\text{seq}}$ for general use, which for BCR sequences is about 0.0025.

We performed an independent $\tau$ optimization for `aa-lbi` and `aa-lbr` (results not shown), which showed both that they are much less sensitive to $\tau$ than are their nucleotide analogs, and that their optima are at comparable values to those of their nucleotide versions. We thus use the same $\tau$ values for both amino acid and nucleotide versions.

As mentioned above, because sequence multiplicity is experimentally difficult to accurately measure, we do not generally recommend its use, since any spurious multiplicities could easily overwhelm the information from unique sequences, which is likely to be more accurate. However, in cases where they are reliable, multiplicities would be an extremely useful source of additional information. If a cell has three sampled offspring, for instance, that is a strong indication of fitness regardless of whether the offspring all have identical BCR sequences. Our implementation of `nuc-lbi` thus incorporates any multiplicity information that has been passed in for each node (see https://git.io/JJCGe). It works by adding additional dummy branches above any node that has multiplicity greater than 1. For instance a node with multiplicity 3 will have 3 branches connecting it to its parent, rather than 1. This represents the case where the three observed sequences all "split off" at the top of the branch to its parent. In reality a split point somewhere in the middle of the branch would likely be more accurate, but we think that the current approach is a reasonable first approximation.

**`nuc-lbi` normalization.** In general, both the absolute magnitude of `nuc-lbi` and its value relative to other sequences are meaningful. For influenza virus evolution, however, which was the case of most interest to [5], only the relative value is useful, since there is only one global influenza "clonal family", and we know that at least some of these viruses will survive in the future (i.e. we definitely want to "choose" some of them). They thus normalize `nuc-lbi` to the maximum value within the tree [70], discarding all information on its

magnitude. Our case, however, is quite different: we are trying to determine how good an antibody is likely to be, so we care very much about the magnitude. Magnitude tells us whether this is a really branchy bit of the tree, not just whether it's branchier than the rest of the tree.

We go further than using the magnitude of `nuc-lbi`, however, and normalize this value relative to two universal and intuitively meaningful minimum and maximum values. One reason is that raw `nuc-lbi` can only be compared to other values that were calculated with the same $\tau$: a comparison to calculations that used a different $\tau$, or simply don't report $\tau$, is meaningless.

We thus look for some theoretically meaningful minimum and maximum values of `nuc-lbi`, and set them to 0 and 1. Note that these don't have to be the actual smallest and largest possible values in order to be useful: for instance the Centigrade temperature scale definitions of 0 and 100 are entirely analogous.

To find a maximum value, we construct a synthetic tree representing the "very branchy" case, then search among its nodes for the maximum value. We generate such a tree by bifurcating after every point mutation (i.e. after a branch length of $1/\ell_{seq}$), then find the node in the tree with maximum `nuc-lbi`. In order to avoid dependence on the depth of this tree, we start with small N generations, and increase depth until (hopefully) reaching an asymptote. If it diverges, i.e. does not reach an asymptote, the exercise is not meaningful (indeed when we N-furcate for $N > 2$ it never converges, results not shown). Calculating this numerically, we find that the maximum value converges to an asymptote for $\tau$ less than around $1/\ell_{seq}$, but diverges for larger values (S10 Fig). Luckily $\tau$ optimization resulted in a comparable recommendation. While this bifurcating tree is of course only one of many possible choices, its purpose is to serve as some reasonable benchmark: when looking at a node with `nuc-lbi` of 1, it's very useful to know that its local area is roughly as branchy as a tree that bifurcates every $1/\ell_{seq}$.

Finding a minimum value is easier: we construct the (deep) caterpillar tree and find the minimum `nuc-lbi` among its nodes. This minimum is simply $\tau$, a result which can also be obtained by performing a trivial analytic integration. We thus normalize `nuc-lbi` such that this minimum value is 0, and the maximum value (previous paragraph) is 1. Note that leaves in shallow trees can thus have values less than 0, and dense trees can result in values greater than 1.

**Dummy branches.** An inherent bias in the `nuc-lbi` calculation is that nodes near the edge of the tree (i.e. near root or leaves) are systematically biased low. While there is no way to address the underlying reason for this—by definition we don't know what happens before root or after leaves—we can test some corrections. Calculating `nuc-lbi` on a (normal) tree that ends abruptly at leaves and root amounts to using the implausible prior that root sprang forth from nothing, and that every leaf died immediately upon sampling (or rather, would have died even without being sampled). We can instead use the much more plausible prior that the status quo at these times continued to ±infinite time by adding long "dummy" branches above root and below each leaf. Unfortunately this does not turn out to improve performance (results not shown). We suspect that this is because we actually know another key piece of information that is not encoded in `nuc-lbi`: most novel mutations are deleterious. In other words the fact that `nuc-lbi` is biased low for leaves is compensated for by the fact that for external reasons we know that leaves are typically low fitness.

**Relative fitness.** We also note that, by design, `nuc-lbi` measures a cell's fitness relative to cells alive at the same time (i.e. cells against which it was actually competing). Thus in any situations that involve sampling sequences from many different time points (e.g. sampling lots of ancestral sequences) performance suffers in an absolute sense. This can be ameliorated in

practice simply by paying attention to the location of nodes within a tree, and keeping in mind that affinity likely increases for much of the length of the tree.

### AA local branching index (`aa-lbi`)

The basic goal of this metric is to recreate the local branching calculation on a tree that reflects only nonsynonymous changes, thus ignoring the significant noise introduced in nucleotide `nuc-lbi` by synonymous mutations. To make this "amino acid tree", we begin with the nucleotide tree, and set the length of each edge to the fractional amino acid hamming distance between the sequences of its two nodes. Unlike nucleotide `nuc-lbi`, this procedure requires an inferred ancestral sequence for each internal node, so it cannot be run on trees produced by faster methods such as FastTree that are suitable for use on full repertoires, and instead requires methods such as RAxML (see "Tree inference methods" below). Another possibility would be to infer the tree directly on the amino acid sequences; while this would result in a less accurate topology, it might be accurate enough for the `aa-lbi` calculation.

### local branching ratio (`nuc-lbr`)

We calculate `nuc-lbr` with the same integrals as `nuc-lbi`, so much of that discussion applies here as well. The difference is that instead of adding up the branch lengths in all directions, we instead divide those below the node by those above and beside it (S1 Fig). This results in a very sharp distinction between the effects of mutations above the parent vs those below.

As for `nuc-lbi`, we performed an optimization of the decay length $\tau$, but with quite different results (S9 Fig). First, the variation in performance is much more significant. Second, there is no peak in performance: instead it improves monotonically as $\tau$ increases to very large values. This makes sense since larger $\tau$ means that more of the tree is visible, which adds information. For `nuc-lbi`, however, there is a countervailing effect: resolution as to which node we're focusing on is also determined by $\tau$, which disfavors large $\tau$. But the resolution of `nuc-lbr` is determined by the sharp distinction between numerator and denominator, so $\tau$ can increase without penalty. Another way of explaining why `nuc-lbr` improves with larger $\tau$ is that if a fitness-increasing mutation has just occurred, the chances of having many offspring in the first generation is likely not very close to 1 even if it is strongly advantageous (for example, let's say it's 1/3). But as we proceed down the tree, the chance of any of the next few generations having many offspring is much closer to one, for instance after five generations it would be $1 - (1/3)^5 = 0.99$. We thus (somewhat arbitrarily) choose to use $\tau$ of $20 \times 1/\ell_{seq}$ for `nuc-lbr`.

While the same normalization procedure could be followed as for `nuc-lbi`, there is no point, because `nuc-lbr` is an inherently unitless ratio.

In the case of `nuc-lbi` above, we concluded that adding "dummy" branches to correct for biases did not make sense. For `nuc-lbr`, however, we have an additional consideration: `nuc-lbr` begins to diverge for nodes near the root, since there is no branch length above the root (in the denominator). Because this is highly undesirable, we add a very long dummy branch above the root node during `nuc-lbr` calculation.

### Tree inference methods

When evaluating the simulation performance of metrics that depend on trees (`nuc-lbi`, `nuc-lbr`, `aa-lbi`, `aa-lbr`, and $\Delta$-`nuc-lbi`), we use only the true simulated tree. This is because we want to separate the ability of the metrics to predict affinity or $\Delta$-affinity from the performance of a particular phylogenetic inference method. We then separately evaluate the robustness of the `nuc-lbi` and `nuc-lbr` calculations to use of the fast but relatively

inaccurate tree inference that `partis` uses by default (S8 Fig). We find in this test that tree inference matters very little for `nuc-lbi`, but is important for `nuc-lbr`. This makes sense, since `nuc-lbi` depends on shorter range comparisons between similar sequences that are likely easier to capture with a heuristic tree method, while `nuc-lbr` is sensitive to the longer range details of ancestral inference. It would be useful to also make this comparison for other phylogenetic inference programs and over the huge variety of tree characteristics encoded by our simulation parameters. As these programs get more accurate, however, they also get much slower, so this will require a substantial time investment.

The cases we can envision where better phylogenetic inference is important will usually involve only a few families, and trees can thus be inferred separately using programs such as linearham [39] or RAxML [40]. These trees would then be passed to `partis` for selection metric calculation using the `--treefname` option of the `get-selection-metrics` action (see https://git.io/JfeGk). Most commonly this would be necessary after an antibody of interest has already been chosen. In such a case we would want to infer ancestral sequences in that antibody's family (which `partis` cannot currently do), and then include these ancestors in the selection metric calculation. Such a workflow was followed in [41] (albeit without the selection metric calculation). This used the `linearham` package for accurate Bayesian inference of both trees and naive sequences (see https://github.com/matsengrp/linearham), and the Olmsted package for visualization (see https://github.com/matsengrp/olmsted), which are both highly recommended.

The fast but heuristic tree inference method that is currently included in `partis` combines the history of its clustering algorithm with the FastTree inference program. Because this clustering proceeds via hierarchical agglomeration [1], the history of the clustering process itself constitutes a tree. While this tree is based only on sequence similarity measures (either inferred naive hamming distance or the `partis` VDJ rearrangement likelihood) rather than a model of sequence evolution, in many cases the result will be quite similar. The biggest inaccuracy in using this approach for tree inference is that the agglomeration frequently merges many clusters together in a single step, resulting in a large multifurcation in the tree. Typically the largest of these happens in the initial clustering step, when it merges together input sequences with very similar inferred naive sequences (for details and thresholds see [1]). We thus refine the clustering-based tree by removing any subtree whose root has more than two offspring, and replace it with a subtree inferred by FastTree [77]. Because FastTree forces any observed ancestral sequences to be very short leaves hanging off of the corresponding internal node, we then also collapse any such leaf with length less than $0.5 \times 1/\ell_{\text{seq}}$. Note that this method does not allow for ancestral sequence inference, which for detailed studies of single families will be quite important.

The speed of this calculation is entirely dependent on how much of the tree needs to be inferred via FastTree, since the non-FastTree parts come along for free from the already-completed clustering. But on typical BCR NGS samples, the only appreciable time is taken for the largest few families, and a family of for instance several thousand sequences can be expected to take a few minutes.

## Decision tree regression (`dtr`)

The fact that we have several very different metrics that perform adequately suggests that we combine them using some form of machine learning. We focused on decision tree regressors, and found that gradient boosting generally performed better than other methods. We tried many combinations of input variables, but found no benefit to reducing their number from the full set, which is shown in Table 3. We trained different `dtr` versions both for choosing

**Table 3. Decision tree regression (`dtr`) input variables.** The among-families `dtr` uses both per-sequence (top) and per-family (bottom) metrics; while the within-family `dtr` uses only the former. `aa-lbi` and `aa-lbr` were not included only because they were developed after completion of the `dtr` study.

| per-sequence metrics | description |
|---|---|
| `aa-cdist` | see Table 1 |
| `nuc-cdist` | see Table 1 |
| `nuc-lbi` | see Table 1 |
| `nuc-lbr` | see Table 1 |
| `n-shm` | see Table 1 |
| `n-shm-aa` | amino acid distance to family naive sequence |
| edge dist. | min. distance in tree to either root or tip |
| **per-family metrics** | description |
| Fay/Wu H | measures selection via excess in site frequency spectrum [68] |
| nuc cons seq SHM | nucleotide distance from family consensus sequence to naive sequence |
| aa cons seq SHM | amino acid distance from family consensus sequence to naive sequence |
| mean `n-shm` | mean number of nucleotide mutations among sequences in the family |
| max lbi | maximum `nuc-lbi` value in the family |
| max lbr | maximum `nuc-lbr` value in the family |

among all families and for choosing within each family, as well as for predicting both affinity and Δ-affinity.

Because we need the `dtr` to perform well for all possible combinations of parameters, we must construct a training sample consisting of families exhibiting a huge variety of characteristics, which requires an extremely large number of families. The types of samples used for the parameter scans, such as Fig 1, are useless for this, since they hold all but one or two parameters constant. We thus made several highly variable samples (Table 4), each generated by first choosing a distribution for each parameter. To simulate a family, a value for each parameter is then drawn from that parameter's distribution. We generated between two and five independent samples for each set of parameter distributions, i.e. the `dtr` was trained on one sample of the indicated size, while there was at least one other sample for testing that had identical underlying parameters, but was generated with a different random seed. We also tested on all other combinations of parameter distributions, as well as the slice/scan samples from e.g. Fig 1. Finally, we also tested a comprehensive variety of the parameters describing `dtr` training (e.g. number of estimators, decision tree depth, and pruning details). Note that we test on training samples only in order to evaluate statistical overtraining, i.e. to ensure that performance is comparable on samples that differ from the training sample only by random seed; we never report or talk about the "performance" on a training sample.

Here we summarize our conclusions; full results can be found at https://zenodo.org/record/3929565. We managed to create a `dtr` that in most regions of parameter space roughly

**Table 4. Parameters used for `dtr` training samples.** Versions v0-v2 sampled parameters for each family from uniform distributions of the indicated mean and half-width, while v3 sampled with equal probability from the indicated discrete values.

| label | N families per sample | N samples | carry cap. | obs times | N sampled | selection strength |
|---|---|---|---|---|---|---|
| v0 | 1000 | 5 | 1500 ± 1000 | 150 ± 75 | 150 ± 100 | 0.75 ± 0.25 |
| v1 | 50000 | 5 | 1500 ± 1000 | 150 ± 75 | 30 ± 7.5 | 0.75 ± 0.25 |
| v2 | 300000 | 2 | 1500 ± 1000 | 150 ± 75 | 20 ± 7.5 | 0.75 ± 0.25 |
| v3 | 50000 | 2 | 250,500,900,1000,1100,1500,5000 | 75,100,150,200,1000 | 15,30,75,150,500 | 0.5,0.9,0.95,1.0 |

recapitulates the performance of the best single metric (usually `aa-cdist` or `aa-lbi`), but improves upon it only slightly (perhaps a few percent). Because of the significant complication introduced by the use of any machine learning method, and their lack of interpretability, we think they are only worthwhile in cases where they improve performance by much more than this. This is of course influenced by the fact that we have one metric (`aa-cdist`) performing well everywhere; if different regions of parameter space required different single metrics, the `dtr` would be much more attractive.

It is difficult, if not impossible, to definitively attribute the `dtr`'s failure to improve performance as much as we expected, since we cannot directly measure whether these expectations were reasonable. However, we think it is highly likely that there is significant additional information in the more poorly-performing variables, and we are also quite confident that a number of features of our use case make it particularly challenging for a machine learning approach. The metrics `nuc-lbi` and `aa-cdist` are far from perfectly correlated (Fig 4, at bottom), and while this discordance is of course not entirely due to useful independent information, `nuc-lbi` clearly contains tree and locality information that is not included in `aa-cdist`. One challenging feature of our case is that the inferred tree quantities that serve as input variables (e.g. `n-shm`, Fay/Wu H) are very noisy predictors of the true tree parameters are (e.g. observation time, selection strength, carrying capacity). To take one example, the relative performance of `nuc-lbi` and `n-shm` completely reverses between low and high selection strength (blue and orange in bottom right of S2 Fig), but Fay/Wu H is far too poor an estimator of selection strength to give the `dtr` an accurate idea where along the x axis it is for a given family (results not shown). It is likely that more sophisticated tree inference would improve performance by allowing the use of more fundamental tree variables as input. It is also of course entirely possible that a different machine learning approach would be able to improve on our efforts; however given the exemplary performance of `aa-cdist` alone, we do not feel that this is a high priority.

## Supporting information

**S1 Fig. Cartoon showing calculation of `nuc-lbi` (left) and `nuc-lbr` (right).** The darkness of each branch represents the exponentially decaying weight factor, which decreases with distance from the node (in red) for which we're calculating the metric. On the left, we show a node with low `nuc-lbi` (top) and high `nuc-lbi` (bottom). At right, in calculating `nuc-lbr` for the red node, we split the tree into two pieces: offspring of the node in the numerator; and parents, siblings, cousins, and their offspring in the denominator. (TIFF)

**S2 Fig. Simulation performance for affinity prediction within families, similar to Fig 1 but for scans across a variety of different parameters.** Performance is shown vs observation times (units of N generations), where sampling occurred at five different time points spanning the indicated values for carrying capacity of both 350 (top left) and 2000 (top right). This is in contrast to Fig 1, where sequences were sampled only at the same, final time point. Performance is also shown vs carrying capacity with 30 (middle left) and 150 (middle right) sampled sequences per family; vs observation time for a non-default affinity calculation utilizing BLOSUM matrices (bottom left); and vs a parameter describing the strength of selection (bottom right). The corresponding among-family plots, as well as plots for many other parameter combinations, are in https://zenodo.org/record/3929565. (TIFF)

**S3 Fig. Simulation performance for within-family affinity prediction with different sampling schemes vs N sampled sequences per family for `aa-cdist` (left) and `nuc-lbi` (right).** Schemes shown are "uniform random" (the default, which is shown in all other plots), "affinity biased" (the probability of sampling each cell is proportional to its affinity), and "perfect affinity" (sample the N cells with highest affinity). Corresponding among-families plots, and plots for all other metrics, are at https://zenodo.org/record/3929565.
(TIFF)

**S4 Fig. Simulation performance for within-family affinity prediction with different numbers of "target sequences" vs observation time for `aa-cdist` (left) and `nuc-lbi` (right).** The target sequence represents a hypothetical optimal antibody toward which selection is directing the cells (see text). Shown for 1, 2, and 4 independently-chosen target sequences (top); as well as for 4, 8, and 16 target sequences divided among the indicated number of "clusters" of target sequences (bottom). Corresponding among-families plots, and plots for all other metrics, are at https://zenodo.org/record/3929565.
(TIFF)

**S5 Fig. Simulation performance for affinity prediction when parameters vary between the families in a sample, shown both for choosing within each family (left) and among all families (right).** Within each plot, families in samples used to calculate points with x values to the left of the dashed line all have the same parameters, whereas those to the right have values sampled from the indicated range. For instance when varying N sampled sequences (top row), 15 sequences were sampled from every family in the leftmost points; whereas in making the rightmost points the number of sequences sampled from each family varied between 15 and 500. In the top three rows, we vary only one parameter at a time between families (N sampled, observation time, and selection strength), while in the bottom row we vary all three at once.
(TIFF)

**S6 Fig. Performance on real data for `nuc-lbi` from [32] (top) and [33] (bottom).** See caption to Fig 3.
(TIFF)

**S7 Fig. Accuracy of the consensus calculation as a function of number of sampled sequences for nucleotide (top) and amino acid (bottom) consensus sequences.** This shows the two extremes of parameters that seem to affect accuracy the most: very early times and small carrying capacities (left) vs very late times and large carrying capacities (right). Each point is the mean ($\pm$ standard error) of 50 families, each with $\simeq 150$ sequences. The y value is the hamming distance (i.e. inaccuracy) between the consensus sequence calculated with the indicated number of sampled sequences (x axis) and the consensus sequence calculated on the entire family.
(TIFF)

**S8 Fig. Comparison of `nuc-lbi` (left) and `nuc-lbr` (right) calculated on true trees (x axis) vs inferred trees (y axis), as scatter plots with Pearson's linear correlation coefficient.** Inferred trees are made with the approximate, but very fast, method run by default `partis` (see Methods). The lefthand plot suggests that `nuc-lbi` is largely insensitive even to very heuristic tree inference. In the righthand plot, on the other hand, the handful of points with highly discrepant true and inferred values indicate that for `nuc-lbr` it is worth using a more sophisticated phylogenetic inference program if at all possible.
(TIFF)

**S9 Fig. Effect of exponential decay length $\tau$ variation on performance when sampling different fractions of the carrying capacity (colors) for `nuc-lbi` (top) and `nuc-lbr` (bottom) when choosing among all families (left) and within each family (right).** Dashed red line corresponds to the value expected from dimensional analysis $1/\ell_{seq} = 1/400$. The fraction observed corresponds to sampling between 30 and 200 sequences from a carrying capacity of 1000. Note that the vertical ordering of lines (i.e. whether performance is better for higher or lower sampling fractions) is not really informative in this plot—the order reverses depending on whether we sample ancestors or not, i.e. to a large extent it just measures the fraction of sampled sequences that are leaves.
(TIFF)

**S10 Fig. Finding a maximum `nuc-lbi` value to use for normalization at different $\tau$ values (colors) with both linear (left) and log (right) scales.** Plots show the maximum `nuc-lbi` value among the nodes in a particular reference synthetic "super branchy" tree as a function of tree depth (N generations). The asymptotic value for $1/\ell_{seq} = 1/400$ is shown in dashed red; the maximum `nuc-lbi` value converges to an asymptote for $\tau$ less than this, while for $\tau$ greater than this the maximum diverges.
(TIFF)

## Acknowledgments

We would like to thank Laura Doepker, Mackenzie Shipley, and Julie Overbaugh for collaborations that motivated this work, as well as Kristian Davidsen and Will DeWitt for helpful discussions. We are indebted to Peter Ralph for invaluable help in editing the manuscript.

## Author Contributions

**Conceptualization:** Duncan K. Ralph, Frederick A. Matsen, IV.

**Data curation:** Duncan K. Ralph.

**Formal analysis:** Duncan K. Ralph.

**Funding acquisition:** Duncan K. Ralph, Frederick A. Matsen, IV.

**Methodology:** Duncan K. Ralph, Frederick A. Matsen, IV.

**Software:** Duncan K. Ralph.

**Supervision:** Frederick A. Matsen, IV.

**Validation:** Duncan K. Ralph.

**Visualization:** Duncan K. Ralph.

**Writing – original draft:** Duncan K. Ralph.

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
