## [Decision Letter · Decision Letter 0]

10 Jun 2020

Dear Dr Ralph,

Thank you very much for submitting your manuscript "Using B cell receptor lineage structures to predict affinity" for consideration at PLOS Computational Biology.

As with all papers reviewed by the journal, your manuscript was reviewed by members of the editorial board and by several independent reviewers. In light of the reviews (below this email), we would like to invite the resubmission of a significantly-revised version that takes into account the reviewers' comments.

We cannot make any decision about publication until we have seen the revised manuscript and your response to the reviewers' comments. Your revised manuscript is also likely to be sent to reviewers for further evaluation.

Sincerely,

Anders Wallqvist

Associate Editor

PLOS Computational Biology

Jason Haugh

Deputy Editor

PLOS Computational Biology

Reviewer's Responses to Questions

**Comments to the Authors:**

Reviewer #1: This manuscript by Ralph and Matsen, tests several possibilities for bioinformatically identifying the “best” antibody from a B cell lineage and likely affinity-enhancing mutations from sequence data alone. This is a crucial open problem in the field, as it has become relatively simple and to generate large quantities of sequencing data, but it remains expensive and time-consuming to synthesize antibodies and test their biochemical properties. Successful prioritization of resources requires the prediction both of the antigen bound by a particular antibody lineage and the relative affinity of the individual sequences within those lineages. The current study addresses the second half of that problem and represents a major step forward in that area.

Most of the metrics the authors test have been previously described in the literature, but none have been systematically investigated or benchmarked against each other. Here, the authors extend an affinity maturation simulation framework that they have previously developed in order to generate a wide range of scenarios in which the various statistics can be tested. Using these data, they show that the Hamming distance between the amino acid sequence of an antibody and the consensus amino acid sequence for that lineage is a good predictor of affinity and that a metric they term the “local branching ratio” can identify branch points on a phylogenetic tree that correspond to the introduction of affinity-enhancing mutations. They also explored the impacts of varying an extensive number of parameters in the simulation to verify that these results are robust to those changes. Finally, they test these metrics on two experimental datasets.

Overall, this manuscript represents a solid methodological advance that will greatly assist the wide range of scientists engaged in antibody discovery. It will also serve as an excellent starting point for further advances that can refine these metrics or combine them with other tools capable of predicting an antibody’s specificity.

Major Comments:

1. I don’t think the “choosing among all families” scenario has been adequately described. What is the intended use case here? The authors seem to contemplate (line 60) selecting from an enriched repertoire, perhaps (line 335) plasmablasts from 7 days after a vaccination. This would certainly be a situation in which even a partially successful metric could be quite valuable. However, the methods only describe the simulation of individual lineage, not of repertoires. Nor do I see systematic testing or parameters involved in repertoire construction the way the authors have done for simulating individual lineages. With each lineage in a different part of parameter space (eg sampling, observation time) that have different, if minor, impacts on the accuracy of aa-cdist, that might add up to even more wobble when trying to compare between lineages. I am especially concerned about widely divergent numbers if input sequences: is the aa-cdist metric really equivalent on a lineage with 5 members sample compared to one with 50? Finally, competition between lineages should correspond to an effective increase in selection strength, which also appears to degrade performance (Figure S2). This is obviously the most challenging part of the already difficult problem that the authors have chosen to address, but I would at least like a little more clarity on what is and isn’t known/possible.

2. It seems that the simulation breaks down at long observation times: Figure S2 shows the aa-cdist metric converging to perfect; Figure S7 shows it as essentially perfect, despite considerable residual variation in the nucleotide sequence. (The left panels of Figure S7 show a more typical relationship between nucleotide and amino acid variation.) The authors explain this with a passing mention of “’optimal’ antibodies” and a citation to reference 20 (lines 304-306). However, they really need to unpack this concept a bit more. Reference 20 explores possible mechanisms for explaining the apparent “affinity ceiling” reached by antibodies during somatic hypermutation, which is not at all the same thing as an “optimal” antibody. The former implies that there are many equally fit sequences (even within a single local maximum), which should result in substantial drift and diversity, possibly even a lack of a strong consensus altogether. The latter, meanwhile, implies stringent negative selection once the “perfect” sequence is reached, allowing only synonymous changes to accumulate (which accounts for the behavior seen in Figure S7). The simulated equivalent of the affinity ceiling discussed in reference 20 would be to treat all sequences with a Hamming distance under some threshold (perhaps 3 or 5) as having equivalent affinity. This would allow the authors to explore the performance of various metrics under this regime of “restricted neutral” evolution. Or, since the authors point out that it’s unclear how often this scenario is biologically relevant, it would be fine to simply acknowledge that their simulations don’t extend to such a situation.

Minor comments:

1. The figures are labeled "shm" instead of "n-shm," though in the text the authors emphasize that these refer to different things.

2. In the legend for Figure S10, it says “The value of 1/l_seq = 1/400 is shown in dashed red,” which is very confusing, as the dashed red line is not drawn at 0.0025 on the y-axis and the line for tau=.0025 is in purple. I would recommend something more like “The asymptotic value for tau = 1/l_seq is shown in dashed red.” In addition it is never stated (here or in the text) what this value of Max_LBI for normalization is.

3. In discussing previous work, it might be worth citing Wang, DeKosky, et al, Nat Biotechnol 2018. It’s a technology for enriching repertoires specific to an antigen of interest and even estimating relative affinity, as well. Certainly it seems like something that could be synergistic with the types of analysis described here.

Reviewer #2: The manuscript by D. Ralph & F.A. Matsen IV describes a methodology to predict the affinity of an antibody (Ab) based on sequence information from a set of evolutionary-related Abs that share a naive ancestor. In addition, the method could detect from an antibody sequence and its inferred ancestral lineage those branches of the tree that are likely to accept mutations leading to an increase in affinity for the antigen.

The work represents an extension of previous efforts by the group to develop theoretical tools for the analysis of next generation sequencing data from B cell repertoires. The research addresses and important area in immunology. However, this reviewer considers that the article needs major revisions. Below are few points that the authors need to address:

Line 209: Fig 1 Legend. The metric used to evaluate the quality of the predictions. i.e., "quantile to perfect", is not defined clearly. How did you computed the numbers “7%” and “12%” for 5 "quantile to perfect” for aa-cdist? Is the description given in Methods section: Evaluation Framework (starting at line 496) relevant, here? How do you define the perfect affinity for the target Abs?

Lines 219-223: The description of the applications of the methodology to antibody repertoires from HIV is insufficient. This reviewer did not find references in the manuscript to the Ab sequences and the experimental data on affinity used for the simulations that may be needed by readers to reproduce the results.

The sentence in Line 219 reads: while there are many papers with both NGS data and affinity information, to our knowledge none of them measure affinity for more than a handful of sequences.

Actually, a number of papers in the literature have reported binding data (Kd), and paired sequence information for set of Abs from few B cell repertoires. The number of Abs in those sets are much larger than those used by the authors, and may represent better tests. Here are two references that may provide better testing sets:

-Goodwin et al. Immunity (2018) 48(2):339-349.e5. Paper describes more than 450 RSV fusion glycoprotein-specific antibodies.

-Bornholdt et al. Science (2016) 351(6277): 1078-1082. Paper describes 349 ebola GP-specific monoclonal antibodies (mAbs).

Ab sequences and binding data are provided in Suppl. Materials of both papers.

Somatic hypermutation-associated insertions and deletions may occur in 1% to 6% of the antibodies from repertoire. How are these considered in your simulations?

Reviewer #3: Antibody affinity is important problem and the ability to predict affinity from sequence would be an enormous accomplishment. Unfortunately, this paper falls far short of this goal. Although my comments below are rather critical, I strongly encourage the authors to follow their idea through by perhaps combining their method with epitope and paratope prediction methods, docking methods or neutralizing assays.

1. The authors need to define what they mean by “affinity” clearly and at the beginning of the manuscript. At line 78 they write, “Since we focus on B cells we refer to affinity; however these methods are in fact measuring evolutionary fitness, and are thus also of much wider applicability.” However, “fitness” is also not defined.

2. I did not find a definition of fitness in the manuscript, but I can generally imagine that it is related to the probability of surviving and reproducing. I think this should be defined and the authors should try to explain why it is not trivial that the consensus sequence is the most fit. I am not saying that it is trivial, it just doesn’t seem, on its face, to be surprising.

3. Affinity is generally expressed as a dissociation constant (Kd), which can be easily measured. The authors should either show the relationship between measured Kd and their internal definition of fitness or make a convincing argument that such data is unnecessary.

4. It is my experience SHM occurs throughout the V region of the antibody. Mutations far from the CDRs can’t easily be understood in terms of affinity. The authors discuss weighting CDRs differently but do not test this idea. I would recommend testing with antibodies that target known antigens with a known binding mode so that they can assess the degree to which the “important” mutations identified by their method involve physical interaction with antigen.

5. In Fig.3 the authors use experimental data to test whether IC50 agrees with edit distance from the consensus. “we find that on real data both aa-cdist (Fig 3) and lbi (S6 Figure) perform roughly in line with expectations from simulation.” To my eye, the lower figure with 6 data points shows a strong positive correlation (smaller magnitude of aa-dist implies larger 1/IC50); in the upper figure, however, the correlation looks very weak and, if anything, goes in the opposite direction (larger magnitude of aa-dist implies larger 1/IC50). This does not appear to validate their method. I don't know what the solution here is, but simply adding this data to the end of the Results, as if it were an afterthought is not helpful.

6. Lines 8-14 of the abstract seem to belong in the Introduction.

**Have all data underlying the figures and results presented in the manuscript been provided?**

Reviewer #1: Yes

Reviewer #2: Yes

Reviewer #3: Yes

PLOS authors have the option to publish the peer review history of their article (what does this mean?). If published, this will include your full peer review and any attached files.

Reviewer #1: Yes: Chaim A Schramm

Reviewer #2: No

Reviewer #3: No
---

## [Decision Letter · Decision Letter 1]

6 Aug 2020

Dear Dr Ralph,

Thank you very much for submitting your manuscript "Using B cell receptor lineage structures to predict affinity" for consideration at PLOS Computational Biology. As with all papers reviewed by the journal, your manuscript was reviewed by members of the editorial board and by several independent reviewers. The reviewers appreciated the attention to an important topic. Based on the reviews, we are likely to accept this manuscript for publication, providing that you modify the manuscript according to the review recommendations.

I would encourage the Authors to address the 3rd reviewer comments and execute the changes suggested, e.g., "the word “affinity” should not be used in the title of this paper. The word “fitness” seems much better and it should be described in the abstract that *predictions are compared with mathematical simulations*."

Sincerely,

Anders Wallqvist

Associate Editor

PLOS Computational Biology

Jason Haugh

Deputy Editor

PLOS Computational Biology

[LINK]

I would encourage the Authors to address the 3rd reviewer comments and execute the changes suggested, e.g., "the word “affinity” should not be used in the title of this paper. The word “fitness” seems much better and it should be described in the abstract that *predictions are compared with mathematical simulations*."

Reviewer's Responses to Questions

**Comments to the Authors:**

Reviewer #1: The authors have ably responded to all questions and critiques posed in the initial round of review, and I think the manuscript can be accepted basically as is. The only thing I think should be added is an evaluation of aa-lbi on the real data set in Fig S6, since this metric outperforms "vanilla" lbi on the simulated data. However, I do not believe this small addition should necessitate an additional round of review.

A few minor but hopefully helpful suggestions if the authors are interested:

1. Given that the word "perfect" seems to be a source of confusion, perhaps the "quantiles to perfect" metric could be renamed to something like "top N accuracy"? It also may be useful to move the practical interpretation of this metric from the legends for Figures 1 and 3 to the main text where paragraph in which it is introduced.

2. For clarity, perhaps lbi and lbr should be nuc-lbi and nuc-lbr, similar to nuc-cdist? Also, Fig 4 and several supplemental figures refer to 'cons-dist-aa' or 'lb index' instead of the 'aa-cdist' and 'lbi' as used in the text.

3. It would probably be more easily interpretable if the data plots in Fig S5 were not continuous across the vertical line.

4. Was tau re-optimized for aa-lbi or did you just to use 1/l_seq_aa?

Reviewer #2: The authors have responded satisfactorily to the issues raised by this reviewer

Reviewer #3: The authors and I seem to be talking past each other. For me, the key problem is that they claim to predict affinity (in the title of the manuscript); but they do not measure affinity. For this reason, they cannot test their prediction. They argue that experimental data is not available

"Unfortunately, the real data we have been able to obtain from the literature is simply too small to form our primary results."

I understand this problem well and I sympathize. This is why I suggested that they attempt to compute affinity in a biophysical way (e.g. by docking). However, this suggestion was rebuffed by the authors:

"However, we have explicitly limited the scope of these methods to those that use evolutionary information."

The methods they choose to use define Kd based on a simulation that is purely mathematical and takes no physical forces for antigen and antibody interaction into consideration. From these simulations they arrive at a relationship between similarity to the consensus and fitness. Since everything in their simulation is a mathematical construct, what they have discovered is a relationship within this artificial mathematical world. It has nothing to do with measured experiments. It doesn’t even attempt to relate to established models of binding affinity, of which there are many (try docking with cluspro to name just one possibility).

The above criticism does not mean that the work is without merit. It simply means that it is hard for someone grounded in a biophysical definition of antibody-antigen recognition to evaluate or imagine making use of their method.

My conclusion as such a reviewer is that the word “affinity” should not be used in the title of this paper. The word “fitness” seems much better and it should be described in the abstract that *predictions are compared with mathematical simulations*.

**Have all data underlying the figures and results presented in the manuscript been provided?**

Reviewer #1: None

Reviewer #2: Yes

Reviewer #3: Yes

PLOS authors have the option to publish the peer review history of their article (what does this mean?). If published, this will include your full peer review and any attached files.

Reviewer #1: **Yes: **Chaim A Schramm

Reviewer #2: No

Reviewer #3: No
---

## [Editor Report · Decision Letter 2]

30 Aug 2020

Dear Dr Ralph,

We are pleased to inform you that your manuscript 'Using B cell receptor lineage structures to predict affinity' has been provisionally accepted for publication in PLOS Computational Biology.

Best regards,

Anders Wallqvist

Associate Editor

PLOS Computational Biology

Jason Haugh

Deputy Editor

PLOS Computational Biology

---

## [Editor Report · Acceptance letter]

28 Oct 2020

PCOMPBIOL-D-20-00710R2 

Using B cell receptor lineage structures to predict affinity

Dear Dr Ralph,

I am pleased to inform you that your manuscript has been formally accepted for publication in PLOS Computational Biology. Your manuscript is now with our production department and you will be notified of the publication date in due course.

With kind regards,

Nicola Davies
